# Collagen Remodeling along Cancer Progression Providing a Novel Opportunity for Cancer Diagnosis and Treatment

**DOI:** 10.3390/ijms231810509

**Published:** 2022-09-10

**Authors:** Kena Song, Zhangqing Yu, Xiangyang Zu, Guoqiang Li, Zhigang Hu, Yun Xue

**Affiliations:** 1College of Medical Technology and Engineering, Henan University of Science and Technology, Luoyang 471023, China; 2Chongqing Key Laboratory of Environmental Materials and Remediation Technologies, College of Chemical and Environmental Engineering, Chongqing University of Arts and Sciences, Yongchuan, Chongqing 402160, China

**Keywords:** cancer, the extracellular matrix (ECM), collagen remodeling, interaction, mechanism, diagnosis, biomarker, therapeutic target, treatment

## Abstract

The extracellular matrix (ECM) is a significant factor in cancer progression. Collagens, as the main component of the ECM, are greatly remodeled alongside cancer development. More and more studies have confirmed that collagens changed from a barrier to providing assistance in cancer development. In this course, collagens cause remodeling alongside cancer progression, which in turn, promotes cancer development. The interaction between collagens and tumor cells is complex with biochemical and mechanical signals intervention through activating diverse signal pathways. As the mechanism gradually clears, it becomes a new target to find opportunities to diagnose and treat cancer. In this review, we investigated the process of collagen remodeling in cancer progression and discussed the interaction between collagens and cancer cells. Several typical effects associated with collagens were highlighted in the review, such as fibrillation in precancerous lesions, enhancing ECM stiffness, promoting angiogenesis, and guiding invasion. Then, the values of cancer diagnosis and prognosis were focused on. It is worth noting that several generated fragments in serum were reported to be able to be biomarkers for cancer diagnosis and prognosis, which is beneficial for clinic detection. At a glance, a variety of reported biomarkers were summarized. Many collagen-associated targets and drugs have been reported for cancer treatment in recent years. The new targets and related drugs were discussed in the review. The mass data were collected and classified by mechanism. Overall, the interaction of collagens and tumor cells is complicated, in which the mechanisms are not completely clear. A lot of collagen-associated biomarkers are excavated for cancer diagnosis. However, new therapeutic targets and related drugs are almost in clinical trials, with merely a few in clinical applications. So, more efforts are needed in collagens-associated studies and drug development for cancer research and treatment.

## 1. Introduction

Cancer is a serious disease for humans with high morbidity and death. The extracellular matrix (ECM) is a non-ignored factor in cancer progression due to the fact that it is the major component of tumor stroma, playing the roles of the physical scaffold and regulator of cell and tissue function. ECM could not only act as a medium to conduct signals but also elicits biochemical and biophysical signaling to excite cells [1,2]. The interaction between tumor cells and ECM is bidirectional and dynamical, which could reshape the morphology of perimalignant tissue continuously. Recent studies have shown that tumors could directly leverage ECM remodeling to create a microenvironment that promotes tumorigenesis and metastasis. Conversely, many cell behaviors are inspired by transformed ECM, such as adhesion, migration, angiogenesis, and canceration [3,4].

In the complex interaction between ECM and tumor cells, collagens play a significant role involved in multiple actions. Collagens are the major component in ECM, which constitute up to 30%—28 different collagens have been identified. Multiple subtype collagens participate in the construction of both matrices of ECM, i.e., base-membrane and interstitial matrix, and create special ECM compositions in different tissues [5,6]. Recently, many studies reported the abnormal appearance in the cancer progression, including degradation, remodeling, fragmentation settlement, linearization, and fasciculation. It is confirmed that collagens are relevant to the precancerous lesion and cancer progression. Based on the mechanisms of collagens involved in the multiple stages of cancer progression, the value of diagnosis and prognosis is developed. Moreover, it provides opportunities to identify new therapeutic targets for cancer treatment. This review investigated the complex response of collagens in cancer progression and summarized the interaction between collagens and cancer cells. Then, we focused on the diagnosis value of collagen-associated imaging and biomarkers. Finally, we discussed the therapeutic opportunities of targeting collagens for cancer treatment.

## 2. Collagen Is a Significant Concern for Cancer Research Associated with the ECM

### 2.1. Collagens Are the Major Component of ECM

Collagens are a superfamily comprising 28 members characterized by collagen α chains [6,7]. According to the supramolecular organization, collagens are divided into fibrillar collagens and non-fibrillar collagens. Fibrillar collagens occupy 90% of the totality, including types I, II, III, V, XI, XXIV, and XXVII. They show elongated, rod, or banded fibril structures under electron microscopy. In comparison, others are non-fibrillar collagens, which form other types of supramolecular structures. Non-fibrillar collagens are further subdivided into fibril-associated collagens with interrupted triple helices (types of IX, XII, XIV, XVI, XIX, XX, XXI, and XXII), network-forming collagens (types IV, VI, VIII, and X), beaded filament-forming collagens (types VI, XXVI, and XXVIII), anchoring fibrils (type VII) and transmembrane collagens (types XIII, XVII, XXIII, and XXV). Collagens participate in the formation of both forms of ECM, i.e., the base-membrane and interstitial matrix. Fibrillar collagens have a clear structural role of mechanical support and dimensional stability, which could provide three-dimensional frameworks for tissue and organs. As an example, type I collagen, as fibrillar collagen, is the main protein in skin, contributing to the tensile strength of skin [6,8]. Non-fibrillar collagens are also essential to maintain tissue structure; for instance, the type IV collagen network is the main scaffold structure of the base membrane. In addition, non-fibrillar collagens are the key regulators to anchor and organize the ECM meshwork. It has been reported that an anchoring bridge is created between the base membrane and the interstitial matrix by the regulation of type VI collagen [9].

### 2.2. The Observation Methods of Collagens in Research and Clinic

The paramorphia of collagens is a significant signal of many diseases, especially fibrosis and precancerosis. Several imaging modalities were developed for quantitative or qualitative analysis collagens. The coarse collagen fiber could be observed directly under the light field of optical microscopy, the visual scenes of which are shown in Figure 1A. However, the tiny collagen fibrils or soft networks are hard to be identified under optical microscopy. Pathological staining technology offers the assistance to catch high-recognition images under light fields, especially for clinicopathological sections. Of note, immunohistochemistry (IHC) could recognize the types of collagens with the assistance of optical microscopy. Fluorescence immunostaining is a method to mark the collagens with a fluorescence dye, then the marked collagens are highlighted under the excitation light of a fluorescence microscope from the background. This method is suitable for all subtypes of collagens, whether fibrous or not; however, this method is hard to apply in practical applications due to the fact that the fluorescence dye is hard to wash completely from the dense networks, causing significant interference to collagen identification. Collagen fiber is a non-centrosymmetry and high second-order nonlinear coefficient structure that could produce the signal of second-harmonic generation (SHG) under two-photon excitation. This provides an excellent opportunity to image collagens, and SHG imaging has been promoted to be a popular and effective approach to investigate collagens in the laboratory. As an example, SHG imaging is developed to investigate collagen fiber organization, which has a high resolution capable of recognizing faint signals [10]. In fact, SHG imaging of collagens is always used combining with confocal microscopy, which could catch the interior signals of the samples to reconstruct 3D images [11,12]. We took an SHG image of type I collagen fibers in an in vitro experiment using confocal microscopy shown in Figure 1B. Due to the fibrous structure differs from other matrices, the pure reflective mode of the confocal microscope is available to take the fiber signal, although the clarity is worse in contrast to SHG imaging [13,14,15]. Figure 1C exhibited the collagen network structure in an in vitro experiment using the reflective mode of the confocal microscope. Electron microscopy is an instrument manufactured by the principle of electron photons, which replaces light beams and optical lenses with electron beams and electron lenses to take the subtle structure imaged with a very high magnification. Fortunately, collagen fibers could be imaged by the two commonly types of electron microscopy, i.e., scanning electron microscopy (SEM) and transmission electron microscopy (TEM) [16,17,18]. SEM could see the morphology of collagens limited to surface and longitudinal. We took the porous structure by SEM, shown in Figure 1D. TEM could evaluate the cross sections through 3D imaging. Figure 1E is the structure of collagen type I under TEM. Cryo-TEM advanced the TEM technique allowing the sample examined to maintain the frozen-hydrated state and removing the step of heavy-metal staining. The early banding analysis of reconstituted collagen fibrils was performed from cryo-TEM images [19,20,21]. Atomic force microscopy (AFM) is another morphology detection instrument based on a completely different principle, i.e., van der Waals force. AFM could be used to confirm the inner assembly of collagen fibrils. As an example, Figure 1F exhibited the microstructure of a single collagen fibril in a high magnification AFM image. However, AFM could only scan the 3D topographic feature, but not the section structure inside the matrix [22,23,24].

Clinically, medical imaging technology is an important contribution to disease diagnosis and surgical guidance, which is popular due to its non-invasive, including ultrasound, X-ray, computed tomography (CT), and nuclear magnetic resonance imaging (MRI). The medical imaging technology allows the sample to be examined in a close to physiological hydration state without chemical fixation of sectioning. Diffraction pattern signals are inspired by high-intensity X-rays scattering from the arranged collagen molecules and fibrils in the bulk matrix. So, X-ray is available to analysis the average diameter, lateral arrangement, and alignment of collagen fibrils; as an example, corneal ultrastructure is obtained by X-ray with a powerful synchrotron source [25,26]. CT and MRI are able to analyze the collagen fibers quantitatively. Karjalainen et al. used micro-computed tomography to analyze the three-dimensional collagen orientation of the human meniscus posterior horn in health and osteoarthritis [27]. Eder et al. used MRI to evaluate the regional collagen fiber network in the human temporomandibular joint disc [28]. Ultrasound is developed to assess collagen microstructure based on the integrated backscatter coefficient (IBC). Mercado, Karla P. et al. employed IBC as a quantitative ultrasound parameter to detect the quantify spatial variations of collagen fiber density and diameter [29]. Kenton et al. prospectively characterize the collagen organization in the Achilles and patellar tendon [30]. However, IHC still plays an irreplaceable role in the clinic due to its ability to recognize the types of collagens. Pathological detection based on IHC is the gold standard for tumor diagnosis in the clinic.

## 3. Collagen Remodeling Is a Significant Signal in Cancer Progression

### 3.1. Precancerous Lesions

Stromal alterations are the reference precursors to predict the progression of carcinoma, especially collagens, which is the main component of the ECM. The occurrence of fibrosis and the base membrane abnormality, which are mainly involved by collagens, are the important cues orienting to deterioration, having a significant clinical meaning. In precancerous lesions, it is likely to occur due to the abnormal ratio of collagen types, new collagen types secreted, and abnormal molecular structure, causing the abnormal ECM.

The fibrosis of organ tissue is an important cue to draw attention to, which is likely to be a stage of deterioration, such as liver fibrosis, lung fibrosis, oral submucosa fibrosis, and so on. Most hepatocellular carcinoma develops through the progression of chronic liver injury, hepatic inflammation, and fibrosis, so liver fibrosis is a precursor of cancerization. It has been confirmed that high fibrosis index is positively correlated with the risk of hepatocellular carcinoma [31,32]. The configuration of collagen types is changed greatly in the progression of fibrillation. In normal liver, the collagens in ECM are type IV and VI, which are non-fibrillar; however, a great accumulation of fibrillar collagens occurs in the fibrotic liver, such as collagen type I and III [33]. Similarly, idiopathic pulmonary fibrosis is considered to have a high risk of concomitant lung cancer in the clinic. What is worse, patients with idiopathic pulmonary fibrosis have a poor prognosis with a 2–5 year survival time, which is worse than liver fibrosis [34]. Enhancing nodules in post-radiation fibrosis in CT imaging could be an early detecting method of recurrent lung cancer [35]. The deposition of collagens in the interstitium is the direct reason for fibrosis. In early pulmonary fibrosis, collagen type III predominates in the matrix; however, the proportion is gradually replaced by collagen type I along the process of pulmonary fibrosis to the late stage [36]. Oral submucous fibrosis is a precancerous disorder and has a 1.5–15% chance of transforming into a malignant tumor. The characteristics of oral submucous fibrosis are abnormal collagen deposition. In oral submucous fibrosis cells, the collagen synthesis is increased and the ratio of the α1(I) to α2(I) chains of type I collagen is ~3:1 whereas ~2:1 in normal cells [37,38].

The analysis of collagen fibers is significant in predicting cancer. Despotovic et al. caught the SEM images of the perimalignant tissue shown in Figure 2A. They found that the altered organization of collagen fibers was observed at 10 cm and 20 cm away from the malignant tumor. The alignment of collagen fibers is step increased as proximity to the tumor [39]. Wu et al. focused on the base membrane in intraductal carcinoma, a precancerous lesion of invasive ductal carcinoma. They found that the base membrane was distorted and elongated compared with the normal cases (Figure 2B). Several types of gynecological cancers are reported differently in terms of precancerous lesions, such as breast cancer, ovarian cancer, and vulvar cancer. Castor et al. characterized the collagen fibers in preneoplastic lesions compared with normal tissue and squamous carcinoma in vulvar cancer using SHG microscopy. They found that the collagen fibers showed better organization in the normal tissue than in the other two stages. The devise parameters of collagen fibers showed reducing in squamous carcinoma and preneoplastic lesions compared with normal tissue, i.e., quantity, organization, and uniformity; however, no obvious difference was observed between squamous carcinoma and preneoplastic lesions [40]. In cervical precancers, several collagen-associated indicators directly affect the quantitative classification of precancerous stages, including the density and degree of linear arrangement, collagen degradation, and the breakage of collagen cross-links. Zaffar et al. focused on this valuable information by developing a series of studies of the spatial frequencies of collagens for cervical precancer detection [41,42]. The expression of collage IV seriously affects the integrity of the base membrane because collagen IV is the main complement of the base membrane. As a precancerous lesion of squamous cell carcinoma in malignant skin tumors, actinic keratosis has shown the premonition of collagen IV low expression. Hirakawa et al. compared the expression of collagen IV using immunohistochemical in actinic keratosis tissue. The result showed that collagen IV in dysplastic areas of actinic keratosis samples was lower than peri-lesional tissue and no longer continuous [43].

### 3.2. Post-Cancerous

#### 3.2.1. Breaching Base Membrane

In the development of cancer progression, collagens become from a passive barrier resisting cancer cells to an accomplice in promoting the progression. The base membrane is a baffle between tumor cells and normal tissue originally; however, it would be breached at the early stage of carcinogenesis. The main reason is collagen IV, which occupies the major complement of the base membrane, is degraded directly or indirectly by tumor cells. In the degrading progress, matrix metalloproteinase plays an indispensable role, which is secreted by tumor cells or stimulated epithelial or stromal cells. In the subsequent progression of cancer development, degrading collagens remains an effective strategy to create roads to invasion or migration. Yan et al. reported a period that collagens exhibited a significant loss in invasive ductal carcinoma compared to the normal case and precursor lesion [44]. Recently, another cue is revealed that cancer cells could break the base membrane just facilitated by physical forces, which is a completely different manner independent of protease. As evidence, the collagen IV meshwork exhibits a densified structure at the adjacent disruption; on the contrary, the collagen IV scaffold should be decreased under the degradation theory. Piercing filopodia is captured in further studies, which is proved to have a pivotal role in the mechanical response model. The force of push and pull by the contractility of piercing filopodia is considered an explanation of the base membrane non-protease disruption [45,46,47].

#### 3.2.2. Enhancing ECM Stiffness

Enhancing the stiffness of ECM is another strategy to promote tumor cell migration and invasion through the pathway of activating integrins to increase the adhesion between cells and substrate. Castor et al. found that the parameters of collagen fibers present higher in metastatic vulvar cancer patients than in that without metastases [40]. High ECM stiffness is mainly realized by increasing the secretion of fibrous collagens and the deposition of non-fibrous collagens, most notably collagen type I and type IV [48,49]. In medical statistics, ECM stiffness is considered a reason for tumor rise incidence with aging because it is a fact that the aged tissues are stiffer due to containing more aberrant cross-linked collagens [50]. High collagen density in tumors is often closely correlated with poor prognosis; however, the association between collagen density and cancer progression is not completely clear. Recent reports studied it from various perspectives. The most accepted view is that ECM stiffness is closely related to cancer-associated fibroblasts (CAFs) [51,52,53,54]. This viewpoint is supported because CAFs are the main producer of abnormal collagen fibers. Shibata et al. reported that CAFs promote ECM stiffness in response to the signals from yes-associated protein 1 (YAP1) [55]. However, Farhat et al. found that abnormal activation and expression of the Lox family of proteins, a group of extracellular enzymes catalyzing the cross-linking of collagens, would lead to the ECM toward increased rigidity and fibrosis [56].

#### 3.2.3. Orienting the Collagen Fibers

Orienting the collagen fibers is a significant manner of remodeling the ECM by cancer cells. The stress of tumor growth remodels the collagen fibers toward the tumor circumference at the tumor periphery. Those oriented collagen fibers provide a highway to cancer cell invasion directly [57,58,59,60]. Meanwhile, the tension of aligned collagen fiber bunches contributes to ECM stiffness [61]. Many studies reported the high orientation of collagen fibers in the tumor location [13,57,59,62]. The mechanism of collagen fibers orientation in the malignant tumor is still unclear. Many scientists dedicated themselves to this study, and they revealed it is a complex process, maybe participated in by tumor cells, CAFs, mesenchymal cells, and interstitial fluid. Tumor cells remodel the collagen fibers through various manners, including the stress of tumor growth, the pseudopod with contractility, the protease, and so on, which is a complex process with multiple pathways [61,63,64,65,66,67]. The examples are shown below. Ray et al. reported that the traction forces from the directed migration of cancer cell clusters are a mechanism of collagen fiber alignment [68]. Drifka et al., using a pancreatic ductal adenocarcinoma model, found that human pancreatic stellate cells could orchestrate the alignment of collagen fibers, and they further found that the aligned collagen fibers, in turn, enhanced cancer cell migration. Bayer et al. showed that the collagen receptor DDR2 in CAFs reorganizes collagen fibers at the tumor-stromal boundary [69]. Hanley et al. reported that CAFs could induce the formation of elongated collagen fibers [70]. Del Amo et al. constructed a collagen-based osteoblasts model in 3D microfluidic devices, and the data suggested that a high rate of interstitial fluid flow could modify the orientation of collagen fibers [71].

## 4. Remodeled Collagens Assist Cancer Progression

### 4.1. Promoting Angiogenesis

In cancer progression, limited oxygen and nutrients are always insufficient for the rapid growth of cancer cells. Angiogenesis is induced under a poor supply of oxygen and nutrient. The angiogenesis process is always accompanied by the upregulation of collagens and ECM-modifying enzymes, which has been confirmed in different human tumor types. That is because the collagen network provides the scaffold for recruited endothelial cells migrating during angiogenesis. Even the remote collagen fiber network is induced by tumor cells into orientation steers for angiogenesis. We illustrated the process of angiogenesis induced by collagen fibers in Figure 3A. Piotrowski-Daspit et al. took a confocal image of a representative tissue comprised of breast cancer cell MDA-MB-231. They found that the collagen fibers showed high alignment obviously (Figure 3B). The collagen network provides the intact physical connection with tumor tissue for vascular endothelial cells’ mechanical sensing, which is considered essential during angiogenesis [72]. Nathaniel et al. observed the fibril organization around angiogenic sprouts and growing neo-vessels in real time. They found that a strong association of fibrillar collagens occurred during vessel reconstitution and a substantial collagen fibril reorganization at the sites of sprout and neo-vessel tips [73]. Luthria et al. investigated the vasculature around the tumor in the dense collagen micro-environment—shown in Figure 3C [74]. Niels et al. reported that type VIII collagen was elevated in diseases associated with angiogenesis and vascular remodeling, such as pulmonary fibrosis and cancer [75].

Another vasculature-like structure could also support tumors with blood, which is called vasculogenic mimicry constructed by cancer cells without endothelial cells. The primary trigger is dense collagen, inducing cancer cells to form interconnected networks [4]. Velez et al. found that the collagen matrixes with small pores and short fibers induced vasculogenic mimicry. The upregulation of β1-integrin is triggered by the collagen matrix architecture and is considered a significant reason in the follow-up research [77]. The non-fibril collagens contribute to vasculogenic mimicry, such as collagen type IV, VI, and XVI. Bedal et al. found the NC11 domain of human collagen XVI, one non-fibril collagen, induced vasculogenic mimicry in oral squamous cell carcinoma cells. The process is realized by triggering the generation of tubular-like net structures on a laminin-rich matrix [78].

### 4.2. Promoting Invasion

Escaping from the primary site is another strategy employed by cancer cells to obtain supplies; collagens also play a significant role in this process. Collagen fibers induce cancer cell migration by contact guidance [68]. Especially, the aligned collagen fibers are confirmed to provide “a highway” to cancer cell invasion [79,80,81]. The guidance of collagen fibers runs through the major process of cancer progression. In the early stage, the collagen fibers that are perpendicular to the solid tumor provide conduits to escape and penetrate the base membrane [58,82]. In the following process, the interface between the collagen fiber bundle in ECM and the peripheral interstitial components provides the opportunity for breakthrough for tumor cells. We illustrated the intravasation guided by collagen fibers in Figure 4A. The role of the “highway” has been verified by several reports. In an experiment on the orientation of collagen fibers, cancer cells could break through the high density of Matrigel along with the collagen fibers, while they were unable to do that in the absence of oriented collagen fibers—shown in Figure 4B [59]. In the process of intravasation (cancer cell entering blood vessel) and extravasation (cancer cell exuding blood vessel), collagen fibers as important participators are remodeled firstly and, in turn, guide the cancer cell invasion [83,84,85,86]. Moreover, the orientation and the deposition of collagens make the matrix stiff. The stiffness of the ECM could induce the enhancement of transforming growth factor-β (TGF-β) to increase the cell adhesion to the substrate, further promoting the migration and invasion of tumor cells. Masoud et al. reported that ECM stiffness could make a bridge in the base membrane through the related pathway of transforming growth factor-β (TGF-β), which contributed to EMT [87]. Figure 4C illustrates the process of EMT induced by the high stiffness ECM. Han et al. displayed a merged image of the SHG signal and light field of the pathological section (Figure 4D). They found that the tumor cells invaded along with the direction of fibers [59]; however, the mechanism of the stiffness ECM acting on tumor cells is complex; there are other pathways being investigated as well. For instance, Anne et al. investigated the immune modulatory properties of collagens in cancer, and they revealed that collagens could affect the function and phenotype of various types of tumor-infiltrating immune cells, such as tumor-associated macrophages and T cells [88]. Yu et al., from the perspective of drug transport, revealed that ECM stiffness was a barrier to drug screening at the tumor site [89].

Non-fibril collagens also participate in the invasion process. Fang et al. reported that collagen type IV occurred a series of changes to provide a proper tumor microenvironment for cancer invasion. In the initial stage, collagen IV presents an irregular sheath in the base membrane, and then it is degraded and accompanied by linear redeposition to form invasion fronts, which would become the escape sites of cancer cells [90]. Zhang et al. reported that collagen XIII could promote invasive tumor growth, enhance the stemness of cancer cells, and induce anoikis resistance [91]. Karagiannis et al. investigated collagen type XII by immunohistochemistry and found that collagen XII was highly expressed in the invasion front of cancer cells [92]. Kumagai et al. found that the intercellular expression of type XVII collagen could promote collective invasion by producing intercellular adhesion sites for contact following [93]. Miyake et al. reported that collagen IV and collagen XIII played a pivotal role in tumor invasion by inducing tumor budding [94].

## 5. Collagens Provide Opportunities to Cancer Diagnosis and Prognosis

### 5.1. Collagen-Associated Biomarkers for Cancer Diagnosis

Collagens, as the main component of the ECM, would be abnormally expressed or deposited in the location or the pericarcinoma of the cancer site. It is for this reason that collagens are proposed as a diagnostic biomarker in cancer types. Type I collagen is reported to be a good diagnostic marker to detect the metastasis of lung cancer, the expression level of which is a significant indicator of distinct bone involvement happening or not in lung cancer metastasis [95]. The degradation products of type I collagen in serum are a significant reference for diagnosis and prognosis. Nurmenniemi et al. reported that type III collagen N-terminal telopeptide and type I collagen C-terminal telopeptide in serum, which are the degradation products of type I and III collagen, respectively, could be used as a prognostic marker in head and neck squamous cell carcinoma because they are proved closely associated with patient survival [96]. Another report revealed that the levels of matrix metalloproteinase (MMP) generated fragments of type I collagen in serum are valuable as a diagnostic biomarker for lung cancer [97]. The variety of collagens is valuable as a biomarker in the diagnosis of numerous cancers, and we collected and summarized them—we present this in Table 1. Of note, many generated fragments of collagens in serum are available for cancer diagnosis, which provides a convenient detection method through drawing peripheral blood.

### 5.2. The Predictive Value of Collagens

Collagens have high value in monitoring cancer processes, prognosis, and recurrence. The predictive value is exhibited in many types of cancer, including breast cancer, prostate adenocarcinoma, lung cancer, hepatocellular carcinoma, colon cancer, and pancreatic cancer. In breast cancer, the qualitative descriptors of collagens at the boundaries between tumor and stroma are important indicators for tumor staging. In invasive breast cancer, the 5-year disease-free survival in patients with low tumor-stromal ratios is poorer than the patients with high tumor–stroma ratios. In lung cancer, high levels of collagen I in serum and tissue demonstrated a significant decrease in survival. Furthermore, collagen metabolic components and prolyl hydroxylases have been marked as the predictive factors of lung cancer presence, progression, and outcome. Abnormal collagen expression is associated with cancer overall survival (OS). The prognosis value is outstanding in enriched stromal cancers, such as pancreatic cancer and colon cancer. It is confirmed that the alignment of collagens was associated with the patient survival of pancreatic ductal adenocarcinoma (PDAC). Highly aligned collagen fibers report poor prognosis in PDAC, according to clinical statistics. Collagen types I, III, VI, and XI were shown to be associated with the diverse response of pancreatic cancers, such as proliferation, migration, decreasing E-cadherin expression, and cancer-associated fibroblasts [158]. Furthermore, the percent survival with a high level of collagen type I is observably lower than the low level. It was shown that the level of collagen type I had a negative correlation to OS for pancreatic cancer [159]. In colon cancer, the level of collagen expression is the key indicator to predicting the OS and risk, especially the types of COL1A1, COL1A2, COL3A1, COL4A3, and COL4A6. Please refer to reference [158] for more information on the predictive value of collagens.

## 6. Therapeutic Opportunities of Cancer Target to Collagens and the Collagen Associated Molecules

Collagen remodeling plays a significant role in cancer progression, as the content described above. It provides the possibility for therapeutic cancer targeting to collagens undergoing remodeling, including inhibiting the synthesis of collagens and interdicting the reactivator on the signal pathway. Of note, many small molecule inhibitors of collagen synthesis and functioning are available as anti-cancer drugs. They could accurately point to the target, avoiding damage to normal cells, which is an absolute advantage over conventional chemotherapeutic agents.

### 6.1. Inhibiting the Synthesis and Secretion of Collagens

Inhibiting collagen synthesis and secretion is a strategy for cancer therapeutics, which could prevent the series of effects initiated by collagen remodeling. For instance, as a catalyst in a key step of collagen biosynthesis, CP4H was linked with cancer metastasis in recent studies. Undoubtedly, CP4H has confirmed a new target for anti-cancer drugs. Several drugs targeted to CP4H have been developed as CP4H inhibitors, such as Ethyl 3,4-dihydroxybenzoate (EDHB) and 2-(5-carboxythiazol-2-yl) pyridine-5-carboxylic acid (pythiDC) [160]. Lysyl hydroxylation is a key step for collagen cross-link and deposition, which is a potential target to inhibit collagen synthesis. Aiming at the collagen remodeling induced by aberrant lysyl hydroxylation and collagen cross-link, lysyl hydroxylation is developed as a potential target for cancer therapeutic. Minoxidil, as an inhibitor of lysyl hydroxylation, is confirmed to have anti-invasive effects on human breast cancer [161]. Relatively, procollagen-lysine 2-oxoglutarate 5-dioxygenase (PLOD) drew much attention as the catalyzer of the process of lysyl hydroxylation. Increased PLOD expression has been detected in many types of cancer. Targeting PLODs is considered a potential strategy for cancer treatment; however, there are still no reports revealing the related anti-cancer drugs [162]. More available targets and the related drug aiming at collagen synthesis are summarized in Table 2.

### 6.2. Interdicting the Receptors

The collagen receptors are the direct trigger of the interaction between cancer cells and collagens. Cancer cells sense the surrounding microenvironment by responding to the biochemical and mechanical properties of transmembrane receptors, including integrins and discoidin domain receptors (DDRs). Interdicting the receptors of collagens is another effectual strategy. We summarize the related targets and the corresponding drug below.

#### 6.2.1. Integrins

Integrins are the important receptor of collagens on the cell membrane. It is a large family, at least including 24 different functional heterodimeric receptors distinguished by 18α-subunits and 8β-subunits [190,191]. According to the reports by far, the integrins of α1β1, α2β1, α3β1, α4β1, α5β1, α6β1, α9β1, α10β1, α11β1, α5β3, and α5β8 have been found to be involved in tumor growth and metastasis by the regulation of collagen-binding integrin signal [190,192,193,194,195,196,197,198,199,200]. Further, integrin-mediated pathways are reported many times to connect to drug resistance [201,202]. So, integrins are considered an attractive drug target for cancer therapeutics [203]. This concept is largely encouraged by preclinical studies. Cilengitide, an inhibitor of integrin αvβ3 and αvβv, has been developed as an anti-cancer drug in various tumor types. Several clinical trials were carried out on diverse cancers, such as lung cancer, breast cancer, glioblastoma, prostate cancer, melanoma, and squamous cell cancer [204,205]. Abituzumab (EMD 525797), a monoclonal antibody targeting integrin alpha nu heterodimers, was also demonstrated as an anti-cancer drug. The phase I clinical trial was completed in ovarian cancer patients with liver metastases and the phase II clinical trial was completed in metastatic colorectal cancer; however, successful clinical trials are few in number. For example, cilengitide failed to improve survival for glioblastoma patients in the phase III trial, despite the standard care. At the same time, the trial data of combining abituzumab showed no improvement in the progression-free survival of patients compared with the standard of care alone. So, further verified trials are essential based on stratifying the patient population [206]. Exploiting tumor-specific integrin expression profiles or downstream integrin effectors is also an alternative strategy to target for the development of anti-cancer drugs. Many typical integrin activation factors are noticed, such as FAK, LOX, mucins, and the corresponding inhibitors are developed for cancer therapeutic. Moreover, snake venom disintegrins were confirmed to inhibit integrins and further effected cancer treatment. We summarize the data in Table 3 below.

#### 6.2.2. DDRs

Another important receptor is DDRs family, including DDR1 and DDR2, which have been proved to regulate various cellular signaling pathways, including cell proliferation, adhesion, migration, and matrix remodeling. DDRs are a subfamily of receptor tyrosine kinases, activated by the triple-helical structure of collagens in the interaction. It is revealed that DDRs possess a special activation mechanism, which initiates the pathways leading to autophosphorylation through collagen binding. In the fibrillar collagens, DDR1 and DDR2 could respond to collagen type I, II, III, and V, while in the non-fibrillar collagens, type IV, VI, VIII, and X are also the activator. However, the mechanism of extracellular collagen binding and activation of the cytosolic kinase domain of the receptors is not clear so far. A recognized theory is that DDRs occur dimerization before the ligand binding with collagens, which is different from the other receptor tyrosine kinases undergoing dimerization after ligand binding [210]; then, the amino acid produces residues of collagens as the sites bind with DDRs dimerization [211]. In turn, the activated DDRs could trigger the signal transduction pathways of cell behaviors, such as proliferation, migration, and invasion [212]. In many cancers, overexpression of DDRs is associated with a poor prognosis. Deng et al. demonstrated that collagen-induced DDR1 activation in cancer cells could recruit tumor-associated neutrophils to form extracellular traps, enhancing the subsequent cancer cell invasion and metastasis [213]. The imbalance expression of DDRs has been demonstrated to be associated with most cancers. Huo et al. revealed that high expression of DDR1 was associated with poor prognosis in pancreatic ductal adenocarcinoma [214]. Xie et al. found that overexpression of DDR1 promoted the aggressive growth, migration, and invasion of bladder cancer cells, in which process collagen IV was a signal axis [215]. As DDRs play a significant role in cancer progression, DDRs would be new promising targets for cancer treatment, such as the design of DDR inhibitors for use in clinical settings. Some drugs were developed to inhibit DDR expression, such as dasatinib, imatinib, nilotinib, and ponatinib. Dasatinib was confirmed enabling to inhibit gastric cancer cell migration and invasion in the assays. Several clinical trials were completed in phases I and II, such as lung cancer, breast cancer, and prostate cancer. Nilotinib was reported to reduce metastatic colorectal cancer invasion by inhibiting DDR1 kinase activation. Many clinical trials are currently underway in several types of cancer, such as breast cancer, gastrointestinal stromal tumors, and so on. The antibody–drug conjugate targeting DDRs is utilized in anti-cancer drug development. As an example, T4H11-DM4 is demonstrated to be effective for colon cancer. Many other drugs were developed to target DDRs in recent years—shown in Table 4.

### 6.3. Targeting to Collagen-Induced Chemoresistance

More and more data implicated the desmoplastic reaction is substantially related to chemoresistance in chemotherapeutics. It is reported that the pancreatic cancer cells grown in collagens demonstrated low sensitivity to gemcitabine chemotherapy. A further study showed that three-dimensional collagens enabled an increase in ERK1/2 signaling, which is known to promote chemotherapy resistance in several cancers. In other malignancies, it is repeatedly reported that collagens protect cancer cells against chemotherapy. For instance, in lung cancer models, collagens are shown to provide survival signals to attenuate the effects of chemotherapy. In this regard, MT1-MMP plays a critical physiological role in modulating growth factors and integrin signaling to enhance ERK1/2 phosphorylation in the collagen microenvironment. Moreover, the increasing density of the ECM initiated by collagen fibers and collagen deposition attenuates the permeability of drug delivery in chemotherapy. Targeting collagen-induced chemoresistance is an effective strategy to promote the chemotherapy effect. Some drugs are developed targeting the intermediary in the signal pathway. Moreover, enhancing the collagen penetration of anticancer drugs is an available strategy for cancer therapeutics, which targets collagen-associated stiffness ECM and high dense collagen fibers, increasing the drug efficacy [252]. We summarize the related drugs in Table 5.

## 7. Conclusions and Future Perspectives

Collagens are the main component of the ECM, and their remodeling occurs along with all processes of cancer progression. Collagens occur in abnormal morphology and distribution in precancerous lesions, which seriously affect the topography of the ECM. This provides a chance for clinicians to discover the pathology through medical imaging or pathological for cancer early diagnosis. In cancer development, collagens cause different stages of ECM topography, which contributes to cancer staging. Moreover, collagen-associated biochemical indicators are the significant biomarkers of cancer diagnosis and prognosis. Collagen participation in cancer progression is not only reflected in the remodeling under the influence of tumor cells or tumor-associated cells but is further revealed as the role of guider or inducer for cancer cell invasion. The “highway” for cancer cell invasion and the high stiffness of ECM are all enhanced by collagens, corresponding with the aligned collagen fiber bundle and collagen deposition, respectively. Based on the mechanism of the interaction between collagens and cancer cells, many opportunities for cancer therapeutics are revealed by disturbing or blocking the requirement in the interaction. The targets are diversely located across the pathway of collagen synthesis, binding to receptors, degradation, and drug transport. Overall, collagens provide many opportunities, whether for cancer diagnosis or cancer treatment; however, there are still many challenges from the exploration of therapeutic targets to drug development because there are only a few drugs allowed to enter the clinical application, while most of them are just in clinical trials or waiting for further clinical studies. Several questions and propositions are provided for further research. Firstly, the mechanism of collagen remodeling and its interaction with cancer cells is not completely clear. So further studies are essential on the mechanisms of collagen remodeling and the interaction between collagens and cancer cells. In our opinion, small molecule inhibitors are worth developing as anti-cancer drugs due to their excellent location and potential to cause no damage to normal cells. It would be a potential strategy to block the collagen remodeling; however, it is hard to implement for the reconstructed collagens. So, how to reduce the effectiveness of the reconstructed collagens is a valuable issue. In addition to the targets in the action pathway, we propose that ablating the reconstructed collagens is another strategy for cancer therapeutics. Moreover, to enhance the efficacy, we highlight the significance of finding a suitable nanocarrier to increase the drug transport capacity. Overall, further efforts are urgent in the collagen-associated mechanism and the therapeutic strategy for cancer.

## Figures and Tables

**Figure 1 ijms-23-10509-f001:**
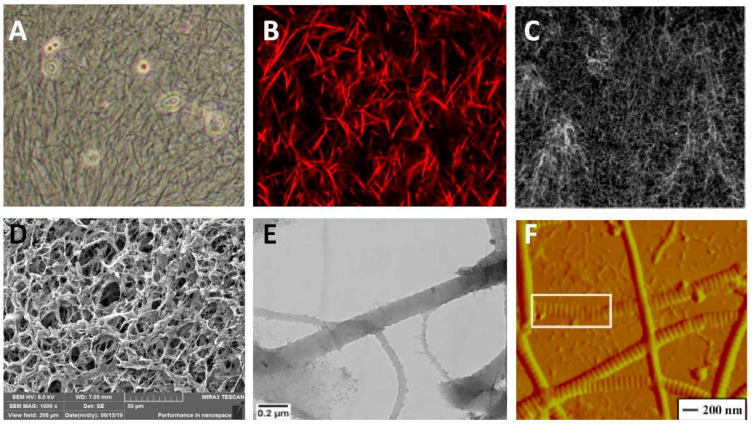
Several examples of collagen detecting methods. (**A**) light field of optical microscopy. (**B**) SHG imaging. (**C**) The reflection mode of confocal microscopy. Reprinted/adapted with permission from Ref. [3]. (**D**) SEM imaging. (**E**) TEM imaging. Reprinted/adapted with permission from Ref. [6]. (**F**) AFM imaging. Reprinted/adapted with permission from Ref. [12].

**Figure 2 ijms-23-10509-f002:**
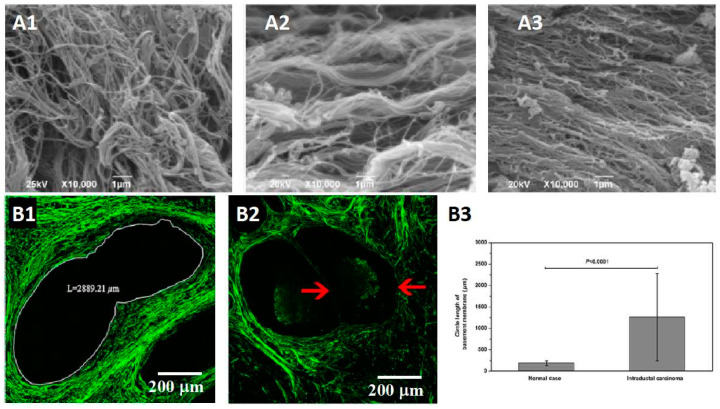
Collagen remodeling in precancerous lesions is a signal of cancerization. (**A1**–**A3**) The SEM images of collagen fibers in the health tissue and 10 cm and 20 cm away from the malignant tumor [39] (**A1**) The collagen fibers in the health tissue are thin collagen fibers forming a dense network. (**A2**) The collagen fibers at 20 cm away from the tumor, thick and aligned. (**A3**) The collagen fibers at 10 cm away from the tumor, highly aligned. (**B1**–**B3**) The difference of base membrane in intraductal carcinoma, a precancerous lesion of invasive ductal carcinoma [44] (**B1**) The base membrane shows a distorted structure with a larger size. (**B2**) The base membrane is destroyed in the evolution from intraductal carcinoma to invasive ductal carcinoma. (**B3**) The statistics data of the circle length of base membrane. The circle length is elongated observably in precancerous lesions compared with normal cases.

**Figure 3 ijms-23-10509-f003:**
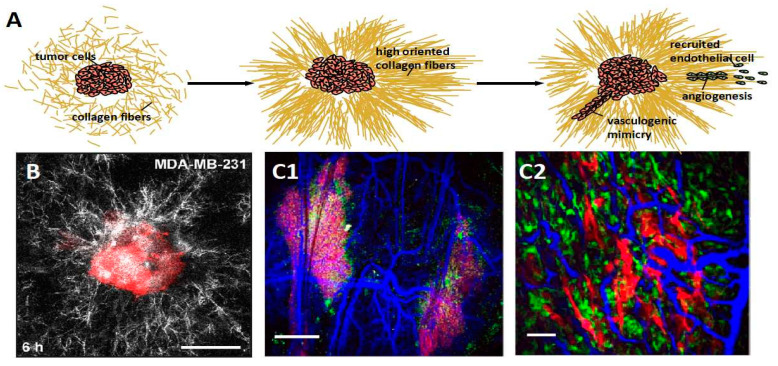
Tumor cells organize the disordered collagen fibers into orientation, further providing a scaffold to recruit endothelial cells (green cell in illustration) for angiogenesis or inducing vasculogenic mimicry constituted by tumor cells (red cell in illustration) without endothelial cells presenting. (**A**) The cartoon diagram displays the process of angiogenesis. (**B**) A confocal image of a representative tissue comprised of breast cancer cell MDA-MB-231. The collagen fibers around the tissue are aligned, obviously. Reprinted/adapted with permission from Ref. [76]. (**C1**,**C2**) The SHG images show the vasculature (blue) formed neighbor tumor within the micro-environment of high dense collagen matrix. Reprinted/adapted with permission from Ref. [74].

**Figure 4 ijms-23-10509-f004:**
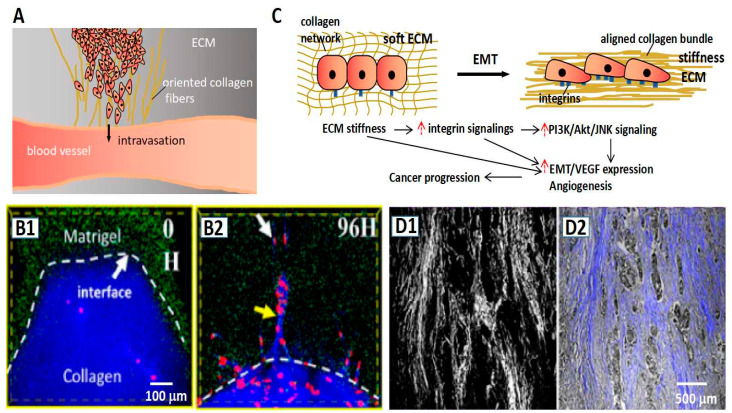
Collagens promote cancer invasion. (**A**) The schematic representation of oriented collagen fibers providing a “highway” to guide cancer cells in realizing the intravasation process. (**B1**,**B2**) The fluorescent images show the stiffness Matrigel (green) region is invaded by cancer cells (red) in the guidance of the oriented collagen fibers (blue) [59]. (**C**) The schematic representation of stiffness ECM inducing EMT accompanied with high adhesion of tumor cells. (**D1**,**D2**) Tumor cells (gray color in (**D2**)) invade along with the direction of collagen fibers. (**D1**) The SHG signal of collagen fibers in the pathological section. (**D2**) The merge of the light field and SHG signal of the pathological section [59].

**Table 1 ijms-23-10509-t001:** The collagen-associated biomarkers and the applicating cancer types in cancer diagnostic.

Collagen Type	Biomarker	Cancer Type	Reference
Type I	the expression level of collagen I	bone metastases of lung cancer; bladder cancer; colorectal cancer	[95,98,99]
	the carboxyl terminal peptide beta-special sequence, total type I procollagen amino terminal propeptide, alkaline phosphatase of collagen I in serum	bone metastases of lung cancer	[100]
	type I collagen C-terminal telopeptide in serum	head and neck squamous cell carcinoma	[96]
	MMP-generated fragments of type I collagen in serum	lung cancer; breast cancer	[97,101]
	the expression level of collagen type I alpha 1	hepatocellular carcinogenesis and metastasis	[102]
	the carboxyterminal collagen type I telopeptid	breast cancer with bone metastases	[103]
	the expression level of collagen type I alpha 2	gastric cancer	[104]
Type III	MMP-generated type III collagen fragment, procollagen type III N-peptide in serum	breast cancer	[101,105]
	the propeptide of type III collagen in serum	pancreatic cancer	[106]
	the level of procollagen III aminoterminal propeptide	malignant head and neck cancer; cervical carcinoma	[107,108]
	procollagen type III N-peptide in serum	gastric cancer; ovarian cancer; lung cancer	[109,110,111]
	type III collagen in serum	hepatocellular carcinoma	[112]
	type III collagen in cancer tissue	malignant pleural mesothelioma	[113]
	the expression of collagen III and collagen III mRNA	Ewing’s sarcoma	[114]
Type IV	the type IV collagen, MMP-generated type IV collagen fragment in serum in serum	breast cancer	[101,105]
	7S domain of type IV collagen in serum	gastric cancer; pancreatic cancer; lung cancer; extrahepatic cancer; hepatocellular carcinoma	[111,115,116,117,118]
	the expression level of collagen IV	malignancy glioma	[119]
	the differential localization of the type IV collagen alpha5/alpha6 chains	colorectal epithelial tumors	[120]
	MMP-degradation of type IV collagen	ovarian cancer	[121]
	the expression level of collagen IV	prostate cancer	[122]
	the expression of collagen type IV	urothelial carcinoma; oral squamous cell carcinoma	[123,124]
	the expression of the type IV collagen alpha1 and alpha2 chains	hepatocarcinogenesis	[125]
	the expression of the type IV collagen alpha1	bladder cancer	[126]
	the expression of collagen IV in tumor location	invasive adenocarcinoma	[127]
	serum N-terminal pro-peptide of type IV collagen 7S domain	hepatocellular carcinoma in patients with liver cirrhosis	[128]
Type V	the expression level of collagen type V alpha 2	colorectal cancer; gastric cancer;	[129,130]
	the expression level of collagen type V	breast cancer; lung cancer	[131,132]
Type VI	MMP-generated type III collagen fragment	breast cancer; colon cancer; gastric cancer; malignant melanoma; lung cancer; ovarian cancer; pancreas cancer; prostate cancer	[133]
	serum collagen type VI alpha 3	pancreatic ductal adenocarcinoma	[134]
	stromal collagen type VI	salivary gland cancer	[135]
	the expression of COL6	pancreatic cancer	[136]
	collagen type VI alpha 1chain	esophageal squamous cell carcinoma	[137]
Type VII	the expression level of type VII collagen	identifying lung cancer subtypes; skin cancer; gastric cancer	[138,139,140]
	the expression level of collagen type VII a 1 chain	clear cell renal cell carcinoma	[141]
Type VIII	the NC1 domain of human type VIII collagen a1 chain in serum	colorectal cancer	[142]
	collagen type VIII alpha 1 chain	breast cancer; colon Adenocarcinoma	[143,144]
Type XI	the expression level of Collagen Type XI Alpha 1 Chain	colorectal cancer; breast carcinoma invasiveness; colon cancer; gastric cancer; pancreatic ductal adenocarcinoma; pancreatic adenocarcinoma; pancreatic cancer	[145,146,147,148,149,150,151]
	the expression of procollagen XI Alpha 1 Chain	breast cancer	[152]
type XIII	the expression of the type XIII collagen alpha1	bladder cancer	[126]
Type XIV	the expression of type XIV collagen	breast cancer	[153]
Type XV	fragments of collagen XV collagen alpha1 in urine	gastrointestinal cancer	[154]
Type XVIII	the expression of type XVIII collagen	pancreatic ductal adenocarcinoma	[155]
	serum endostatin (a fragment of collagen XVIII) levels	colorectal cancer	[156]
Type XXIII	the expression of collagen type XXIII alpha 1 chain	clear cell renal cell carcinoma	[157]

**Table 2 ijms-23-10509-t002:** A summary of drugs targeting the synthesis and secretion of collagens.

Therapeutics Target	Drug	Mechanism	Cancer Type	Status	Reference
collagen type I	Baicalein	inhibiting collagen type I transcription by alleviating TGF-β1 stimulation	lung cancer, osteosarcoma cells, bladder cancer, breast cancer, pancreatic cancer, cervical cancer, oral cancer	a promising candidate awaiting further testing	[163,164,165]
	Phenylbutyrate, sodium phenylbutyrate	as a weak histone deacetylase inhibitor decreasing collagen type I Alpha 1 mRNA transcription	lung cancer, prostate cancer, liver cancer, breast cancer, ovarian cancer, bladder cancer	a promising candidate awaiting further testing	[166,167,168,169,170,171]
	C9	C9 inhibits collagen production by dissociating laribonucleoprotein domain family member 6 (LARP6) from type I collagen 50′SL RNA	-	awaiting further testing	[172]
	Ethyl 3,4-dihydroxybenzoate (EDHB), 2-(5-carboxythiazol-2-yl) pyridine-5-carboxylic acid (pythiDC)	inhibiting collagen synthesis by inhibiting Prolyl 4-hydroxylases (P4Hs), which is a synthesis and regulatory factor of collagen type I	colorectal cancer, breast cancer	awaiting further testing	[160,173,174]
	Minoxidil	inhibiting collagen synthesis by inhibitor lysyl hydroxylases (LHs), which is a synthesis and regulatory factor of collagen type I	prostate cancer, breast cancer, ovarian cancer	a promising candidate awaiting further testing	[161,162,175,176]
	AK-778	inhibiting collagen synthesis by mitigating the interaction between collagen and HSP47, which is a molecule required for collagen type I maturation.	-	a promising candidate awaiting further testing	[177,178]
	CCT365623	decreasing collagen synthesis by inhibiting lysyl oxidase (LOX), which is a regulatory factor for collagen cross-linking	-	a promising candidate awaiting further testing	[179,180]
collagen type XI	LY2157299	inhibiting collagen XI alpha 1 chain (COL11A1) expression by inhibiting the transforming growth factor beta receptor 1 (TβRI)	ovarian cancer, pancreatic cancer, breast cancer	phase II clinical trial	[181,182,183,184,185]
	SC66	as an Akt inhibitor preventing the transcription of COL11A1	colon cancer, ovarian cancer, bladder cancer, lung cancer	under clinical trials	[182,186,187,188,189]
collagen type XI	AK778 and its cleavage product Col003	disrupted collagen binding with the molecular chaperone HSP47 and inhibited collagen secretion	-	awaiting further clinical studies	[177,182]

**Table 3 ijms-23-10509-t003:** Integrin-associated therapy and the related drugs in cancer therapeutics.

Therapeutics Target	Drug	Mechanism	Cancer Type	Status	Reference
integrin	Cilengitide	an inhibitor of integrin ανβ3, ανβ5, α5β1, αIIβ3	lung cancer, breast cancer, glioblastoma, prostate cancer, melanoma, squamous cell cancer	in clinical trials	[203,207,208]
	Abituzumab	inhibiting integrin ανβ1, ανβ3, ανβ5, ανβ6, ανβ8	colorectal cancer. ovarian cancer	in clinical trials	[203,209]
	Etaracizumab	inhibiting integrin ανβ3	melanoma prostate cancer	a phase II trial	[203]
	Intetumumab	inhibiting integrin ανβ1, ανβ3, ανβ5, ανβ6, ανβ8	melanoma prostate cancer	a phase II trial	[203]
	NCT02428270	using a FAK inhibitor in combination with a MEK1 and MEK2 inhibitor	pancreatic cancer	a phase II trial	[206]
	NCT02546531	using a FAK inhibitor in combination with a humanized antibody targeting programmed cell death protein 1 (PD1) and chemotherapy	solid tumors pancreatic cancer	a phase I trial	[206]
	NCT01279603	inhibit MUC1 cytoplasmic tail oligomerization	solid tumors	a phase I trial	[206]
	NCT00565721	as valuable probes in cancer imaging studies to determine both prognosis and treatment efficacy	lung cancer Head & Neck cancer	a phase II trial	[206]
	NCT02683824	an αvβ6 integrin tracer to detect tumors and evaluate treatment response in patients with pancreatic cancer	pancreatic cancer	early phase I trial	[206]
	snake venom disintegrins	inhibiting integrins on transmembrane cellular surfaces	prostate breast cancer lung cancer sarcoma	a promising candidate awaiting further testing	[208]

**Table 4 ijms-23-10509-t004:** DDRs associated therapy and the related drugs in cancer therapeutics.

Therapeutics Target	Drug	Mechanism	Cancer Type	Status	Reference
DDRs	nilotinib	inhibiting the kinase activity of DDR1	colorectal cancer		[216]
	Dasatinib	inhibit DDRs	prostate cancer, glioblastoma, breast cancer, lung cancer, gastric cancer	in clinical trials	[217,218,219,220,221,222,223]
	Nilotinib	inhibit DDRs	colorectal cancer, colon cancer	in clinical trials	[217,224,225,226]
	Imatinib	inhibit DDRs	lung cancer, liver cancer	in clinical trials	[227,228,229,230]
	Ponatinib	inhibit DDRs	lung cancer	in clinical trials	[231,232]
	T4H11-DM4	an antibody-drug conjugate targeting DDR1	colon cancer	awaiting further clinical studies	[233]
	Actinomycin D	an antagonist of the DDR2-collagen interaction	rhabdomyosarcoma, Ewing’s sarcoma, trophoblastic neoplasia, and testicular carcinoma	in clinical application	[234,235]
	LCB 03-0110	inhibiting collagen-induced activation of DDR1 and DDR2 receptors	-	awaiting further clinical studies	[236,237]
	pyrazolo-urea containing compounds 2a, 4a, 4b	inhibit DDR2	-	awaiting further clinical studies	[238]
	7rh	inhibited the kinase activity of DDR1	gastric cancer, nasopharyngeal carcinoma, pancreatic ductal adenocarcinoma, breast cancer, uveal melanoma	in clinical trials	[218,239,240,241,242,243]
	7rj	inhibited the kinase activity of DDR1	-	awaiting further clinical studies	[235]
	DDR1-IN-1	induced a significant inhibitory effect against DDR1	colorectal cancer, lung cancer	awaiting further clinical studies	[232,244,245,246]
	DDR1-IN-2	induced a significant inhibitory effect against DDR1	-	awaiting further clinical studies	[235]
	miR-199a-5p, a targeted delivery of miRNAs	inhibit DDRs	colorectal cancer, renal cancer	awaiting further clinical studies	[247,248,249,250]
	monoclonal antibodies Fab 3E3, 48B3, H-126	inhibit DDRs	ductal breast carcinoma	awaiting further clinical studies	[235,251]

**Table 5 ijms-23-10509-t005:** The drugs that target collagen-induced chemoresistance.

Therapeutics Target	Drug	Mechanism	Cancer Type	Status	Reference
collagen	polyethylene glycol (PEG) & glutaraldehyde co-modified fluorinated chitosan (PGFCS)	as a collagen-targeted transepithelial penetration enhancer creating a tumor-targeted adhesive interface to open the transepithelial-delivery barrier at the tumor site	bladder cancer	awaiting further testing	[253]
	losartan	reducing stromal collagen and hyaluronan production to decompress tumor vessels for enhancing drug delivery.	prostate cancer, colorectal cancer, pancreatic cancer, breast cancer, lung cancer, ovarian cancer, endometrial cancer	in clinical application	[254,255,256,257,258,259,260,261]
Collagen type II	Ivosideni	a selective inhibitor of mutant IDH1, which is a gene mutant site of collagen type II	chondrosarcoma	waiting further clinical studies	[262]
ERK1/2	JaZ-30	downregulates phosphorylation of the extracellular signal-regulated ERK1/2	melanoma	waiting further clinical studies	[263]

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
