# Peer review of "Collagen Remodeling along Cancer Progression Providing a Novel Opportunity for Cancer Diagnosis and Treatment"

_ijms, 2022, doi:10.3390/ijms231810509_

Round 1

Reviewer 1 Report

Dear Authors,

In my opinion, the manuscript may be published after major revision.

The list of the most important comments is as follows:

- In the abstract, the content of lines 11-24 should be shortened and the content of lines 25-31 should be expanded. Then it will fit the purpose of the work, not a duplication of well-known common data.

- For review articles, I prefer more keywords.

- In scientific articles, I do not prefer common statistics on morbidity, that is not the purpose of this paper (lines 36-42).

- I propose to clarify the term collagen, in the abstract term ‘collagens’ is used, in the introduction ‘collagen’. Please assign the correct types. Sometimes, I find it hard to understand the context, if I read the term collagen and only collagen without specifying the type.

- The data given in lines (69-96) do not matter to me. These data are well known. I wrote about it 10 and 20 years ago. They are irrelevant to the purpose of this work.

- The data given in lines 98-160  are not significant. This is a description of the tools used in research. It is not scientific. It does not indicate problems and solutions. The AFM image quality in figure 1F is very poor. Moreover, in my opinion the scale is incorrect. The bar points to 200 nm. Please note that the collagen periodicity is 67 nm.

- The font is incorrect in line 199.

- In the figure 4, a schematic representation of collagen promoting cancer invasion was presented. In journal with such a high IF I expect illustrations based on microscopic study.

- I expect also the use of histological, AFM, SEM images from the literature data to visualize the presented theses.

- Tables should not be randomly broken down by pages.

Best regads,

Reviewer

Reviewer 2 Report

In this paper, Song et al highlighted the potential of assessing collagen remodeling as a biomarker for cancer diagnosis and treatment. The paper is well outlined and written, and the authors discussed each item in detail.

Although the current version of the paper is well articulated, some minor comments can improve the quality of the paper, especially adding more figures in some parts that might confuse the reader. In this regard, adding a few in-slico data derived from publicly available portals showing collagen expression in various cancers and its association with OS is very helpful.

1- In section 2.2, CT and MRI are useful for measuring the density of tissue only; however, the IHC method can reflect the type of collagen, which is not discussed in this part. Moreover, how these remodelling associates directly or indirectly with patient outcomes have to be briefly discussed as well.

2- in page 13, the authors very shortly highlighted the role of collagen on chemoresistance. The authors must first distinguish these events with ECM stiffness, mostly observed in stromal enriched cancer types like pancreatic cancer. Moreover, a huge number of high-profile articles mechanistically show these associations. Due to the importance of these events, the authors should provide more information and enrich this part, which unfortunately is not well discussed compared to other parts.

3- Considering balancing between outlines, based on the title of the paper, it is assumed that most of the paper, not all section be dedicated to the role of collagen in diagnostic and therapy, while the current version of the paper discusses more for mechanisms of collagen remodelling in cancer progression and invasion. So, the Reviewer strongly advises the authors to revise and add more related information.

4- As the aim of developing biomarkers for diagnosis and treatment of cancer is to improve patient outcomes, unfortunately, the authors did not provide any insight into how resolving collagen remodelling helps patient outcomes. How much is worth it when we can treat cancer with small molecule inhibitors rather than chemo agents, meaning that ECM stiffness can not be a strong barrier in the case of small molecule inhibitors, for example, in colon cancer or pancreatic tumours?

5- Finally, it would be valuable if the authors provide some outstanding questions at the end of the paper that require further research.  

Round 2

Reviewer 1 Report

Dear Authors,

I found some issues in the text that need to be clarified:

1.           There should be no reference to tables in the abstract. I propose to change the content of the sentences

“At a glance, a 31 variety of reported biomarkers were summarized in a table” (lines 31, 32)

“The mass data 34 was collected and classified in tables by mechanism” (lines 34, 35)

2.           As keywords I propose to use words and not sentence equivalents, e.g.: the interaction of collagens and cancer cells; mechanism; cancer diagnosis and treatment; collagen associated biomarker

It should be noted that the simpler keywords, the greater the chances of being cited.

3.           I understand that figure 3 on page 10 will be completely deleted.

4.           I suggest that subsections 3.1, 4.1, and 4.2 should begin with the text, not a figure.

5.           In Figures 2 - 5, not all images have bar or magnification information. This needs to be completed.

Reviewer 2 Report

I am happy with the revised verison. It can be accepted. 

Author Response

Dear Reviewer,

    Thanks so much for your affirmation of our work!

kind Regards,

Kena Song